# Minced Cartilage Is a One-Step Cartilage Repair Procedure for Small Defects in the Knee—A Systematic-Review and Meta-Analysis

**DOI:** 10.3390/jpm12111923

**Published:** 2022-11-18

**Authors:** Andreas Frodl, Markus Siegel, Andreas Fuchs, Ferdinand C. Wagner, Hagen Schmal, Kaywan Izadpanah, Tayfun Yilmaz

**Affiliations:** 1Department of Orthopedics and Traumatology, Freiburg University Hospital, 79106 Freiburg, Germany; 2Department of Orthopedic Surgery, University Hospital Odense, Sdr. Boulevard 29, 5000 Odense, Denmark

**Keywords:** cartilage defects, cartilage repair, minced cartilage

## Abstract

Purpose: Approximately 60% of patients undergoing arthroscopy of the knee present with chondral defects. If left untreated, osteochondral lesions can trigger an early onset of osteoarthritis. Many cartilage repair techniques are mainly differentiated in techniques aiming for bone marrow stimulation, or cell-based methods. Cartilage repair can also be categorized in one- and two-stage procedures. Some two-stage procedures come with a high cost for scaffolds, extensive cell-processing, strict regulatory requirements, and limited logistical availability. Minced cartilage, however, is a one-stage procedure delivering promising results in short term follow-up, as noted in recent investigations. However, there is no available literature summarizing or synthesizing clinical data. The purpose of this study was to analyze and synthesize data from the latest literature in a meta-analysis of outcomes after the minced cartilage procedure and to compare its effectiveness to standard repair techniques. Methods: We conducted a systematic review searching the Cochrane, PubMed, and Ovid databases. Inclusion criteria were the modified Coleman methodology Score (mCMS) >60, cartilaginous knee-joint defects, and adult patients. Patient age < 18 years, biomechanical and animal studies were excluded. Relevant articles were reviewed independently by referring to title and abstract. In a systematic review, we compared three studies and 52 patients with a total of 63 lesions. Results: Analysis of Knee Injury and Osteoarthritis Outcome Score (KOOS) sub scores at 12 and 24 months showed a significant score increase in every sub score. Highest mean difference was seen in KOOS sport, lowest in KOOS symptoms (12 month: KOOS sport (Mean difference: 35.35 [28.16, 42.53]; *p* < 0.0001), lowest in KOOS symptoms (Mean difference: 20.12 [15.43, 24.80]; *p* < 0.0001)). A comparison of International Knee Documentation Committee (IKDC ) scores visualized a significant score increase for both time points too ((12 month: pooled total mean: 73.00 ± 14.65; Mean difference: 34.33 [26.84, 41.82]; *p* < 0.00001) (24 month: pooled total mean: 77.64 ± 14.46; mean difference: 35.20 [39.49, 40.92]; *p* < 0.00001)). Conclusion: Due to no need for separate cell-processing, and thanks to being a one-step procedure, minced cartilage is a promising method for cartilage repair in small defect sizes (mean 2.77 cm^2^, range 1.3–4.7 cm^2^). However, the most recent evidence is scarce, and takes only results two years post-surgery into account. Summarized, minced cartilage presents nearly equal short-term improvement of clinical scores (IKDC, KOOS) compared to standard cartilage repair techniques.

## 1. Introduction

Articular cartilage is highly complex tissue that undergoes metabolic changes due to trauma, degeneration, or the deprivation of nutrients through osteochondrosis dissecans (OD) or osteonecrosis [1,2,3,4,5,6]. These factors hinder tissue maintenance and repair mechanisms, leading to loss of cartilage-surface area and ultimately osteoarthrosis. However, there are treatment options to repair and restore articular cartilage surgically. Current interventions can be categorized either as techniques aiming to stimulate bone marrow, or as cell-based cartilage repair. Timewise, differentiation is possible in one- and two-stage procedures depending on whether one or two interventions are needed to integrate cartilaginous material and defect filing [4,7].

Standard procedures include matrix-assisted chondrocyte implantation (MACI) or the osteochondral autograft transfer system (OATS) and osteochondral allograft transplantation (OCA) [3,8]. Current one-stage procedures, e.g., microfracture and OATS, are applicable with good clinical results for smaller defects. Larger defects must often be handled with two-stage procedures (i.e., MACI) [3].

These cell-based procedures are associated with expensive scaffolds and the demanding conditions required for in vitro cell proliferation. Considering cost-effectiveness, a one-stage procedure not requiring demanding in vitro cell proliferation to induce hyaline or hyaline-like repair tissue, would seem more suitable. Minced cartilage implantation is just such a type of cartilage repair; it entails the implantation of autologous cartilage chips [9,10]. Recent investigations have shown promising short-term follow-up results. However, there is a paucity of data on long-term outcomes and comparisons to similar cartilage repair techniques [11]. The purpose of this study was to analyze the most recent literature reporting on outcomes after the minced-cartilage procedure, comparing its effectiveness to established cartilage repair techniques. Hence, minced cartilage is suggested as a reliable alternative in cartilage repair.

## 2. Methods

This descriptive systematic review was conducted according to the Preferred Reporting Items for Systematic Reviews and Meta-analyses checklist guidelines (PRISMA) and submitted to PROSPERO [12,13].

From January 2021 to October 2021, a database search was done independently by the authors (A.F. and T.Y.). MEDLINE, PubMed, Embase, Web of Science, and the Cochrane Library were searched for relevant studies reporting clinical outcome after minced cartilage therapy in patients with cartilage defects of the knee.

### 2.1. Search Strategy


**#**

**Query**

**Results**
1exp Knee Joint/or exp Cartilage, Articular/or exp Cartilage/or exp Orthopedic Procedures/424,8412exp Rehabilitation/mt, su, td [Methods, Surgery, Trends]70,7033exp Cartilage/or exp Hyaline Cartilage/or exp Cartilage, Articular/87,1374exp Treatment Outcome/1,102,95552 or 41,156,74661 and 582,18873 and 660598exp Arthroscopy/ae, mt, rh, st, td [Adverse Effects, Methods, Rehabilitation, Standards, Trends]10,59397 and 857210minced cartilage.mp.1711minced.mp.244612cartilage.mp.98,0871311 and 12651410 or 1365159 or 14637

### 2.2. Eligibility

Our inclusion criteria were: studies between 2000 and 2020 and a minimum follow-up of 12 months. Only publications written in German or English were included. A minimum patient age of 18 years was set to enable comparisons between fully grown adults undergoing the minced cartilage procedure. An evaluation of clinical scores (International Knee Documentation Committee IKDC, Knee Injury and Osteoarthritis Outcome Score KOOS or Lysholm) was mandatory at 12 and 24 months post-surgery. Our exclusion criteria were: an overall modified Coleman Methodology Score (mCMS) <60, follow-up rate <80%, cadaver or biomechanical studies, and animal studies.

Titles and abstracts were screened independently by the reviewers for relevance according to the aforementioned inclusion and exclusion criteria. The full text was obtained to assess the study’s relevance if no abstract was available. We cross-referenced the references within included articles if they had been missed by our search algorithm to make sure we had not overlooked any suitable studies. Appropriate publications were then independently analyzed for the mCMS and risk of bias with ROBINS-I tool [12,13,14,15].

### 2.3. Primary Outcome Criteria

Patient demographics, number of patients, clinical scores (IKDC, KOOS), follow-up period, and surgical technique were extracted by us authors. The hypothesis was, concerning clinical scores (KOOS, IKDC), that the minced cartilage procedure is comparable to those of commonly used cartilage repair techniques.

### 2.4. Statistics

To analyze the collected data, Microsoft Excel (ver. 16.16, Redmond, Wahington, Microsoft Corp., 2019) and Revman5 (Version 5.3. Copenhagen: The Nordic Cochrane Centre, The Cochrane Collaboration, 2014) were used. Given means and standard deviations were pooled using continuous statistic evaluation protocol of RevMan5 and calculated in a forest plot as mean differences for each point of time.

## 3. Results

### 3.1. Study Selection

Our literature search and study selection procedure is depicted in Figure 1. A total of 637 papers were identified by our search algorithm. Moreover, one paper was added from the reference list. These papers were scanned, and any duplicates or topic-unrelated articles excluded. After analyzing the eligibility criteria, three of the five studies were included in our analysis. To acquire the original, complete data set, we contacted the corresponding authors of the papers to be included. Nevertheless, access to complete data was granted in just one case, which hindered our qualitative analysis.

We identified two prospective trails and one randomized controlled trial containing a total 52 patients with a total of 61 cartilage defects at the knee for inclusion in our review.

The number of patients included in these studies ranged from 7 to 25 with a mean age of 30.6 ± 5.4 years.

### 3.2. Risk of Bias Assessment

All included studies possessed evidence-level III or II. Reporting and detection biases are considerable due to the lack of randomization and blinding in two of the three studies. Surgical techniques were reported in detail in every study, minimizing the risk of operational bias even in cases in which several surgeons were operating. To calculate the risk of underlying bias, all included studies were analyzed with the ROBINS-I tool.

Our results for the risk of bias assessment and mCMS scores are shown in Table 1 and Table 2.

### 3.3. Clinical Outcome

In all three studies, the KOOS and IKDC score was used to evaluate clinical outcome. All studies included time points of 0 and 12 months. Farr et al. and Cole et al. also conducted clinical evaluations at 24 months (Table 3). In each study minced cartilage was combined with fibrin glue, which accounts for similar surgical methods and comparable results in respect of treatment methods.

#### 3.3.1. Clinical Scores at 12-Month Post-Surgery

Overall, 105 patients were included in this subgroup analysis. Mean differences for each KOOS sub score have been calculated and displayed in a forest plot. A significant score increase was seen in every sub score. The highest mean difference between pre- and post-surgery score was seen in KOOS sport (Mean difference: 35.35 [28.16, 42.53]; *p* < 0.0001), lowest in KOOS symptoms (Mean difference: 20.12 [15.43, 24.80]; *p* < 0.0001). A detailed analysis of the 12-month score evaluation is presented in Figure 2.

IKDC scores have been reported by Christensen et al. and Cole et al. at 12-month control. Mean difference compared to pre-surgery analysis was significantly higher than pre-surgery (pooled total mean: 73.00 ± 14.65; Mean difference: 34.33 [26.84, 41.82]; *p* < 0.00001) (Figure 3).

#### 3.3.2. Clinical Scores at 24-Month Post-Surgery

Clinical scores for the 24-month time interval have only been reported by Cole et al. and Farr et al. Christensen et al. only reported 12-month results.

A total of 97 patients were thus included for analysis. A significant increase in every sub score was observed.

The KOOS sport subgroup showed the highest increase after 24 months (mean difference: 35.86 [27.21, 44.51]; *p* < 0.00001). Similar to 12 month results, the lowest increase of scores was seen in KOOS symptoms (mean difference: 19.29 [13.11, 25.47], *p* < 0.00001) (Figure 4).

The IKDC at 24 months was not reported by Christensen et al. A significant increase was seen compared to pre-surgery evaluation (pooled total mean: 77.64 ± 14.46; mean difference: 35.85 [20.07, 51.63]; *p* < 0.00001) and is depicted in Figure 5.

## 4. Discussion

As depicted in our overview of how we selected studies, there is very little recent data on the minced cartilage procedure and its long-term results. The studies we included represent the only evidence originating from 24-month follow-ups with satisfactory methodology. Mean defect size ranged between 1.3 and 4.7 cm^2^. For inter-patient comparability, the KOOS and IKDC scores were calculated pre-surgery, and at 12- and 24-month after the minced cartilage procedure. Summarizing the outcomes of this meta-analysis, our study’s key finding is that following minced cartilage procedure clinical scores (KOOS and IKDC) increased significantly at 12 and 24 months compared to pre-surgery measurements.

The latest surgical procedures for cartilage repair can be categorized as one-step or two-step procedures [16]. However, treatment options are limited to the knee’s defect size [17].

The microfracture approach aims to stimulate bone marrow growth by multiple drilling into the subchondral bone, a method by which mesenchymal stem cells and growth factors are emitted from the drilling holes, filling the defect zone with fibrocartilage [18,19]. Compared to hyaline cartilage, which is mainly composed of type II collagen, such fibrocartilage is composed of a mixture of fibrous tissue and cartilage tissue containing type I and type II collagen [20]. Because of the tissue’s fibrosity, fibrocartilage is both less flexible and tougher than hyaline cartilage, resulting in mechanical drawbacks compared to hyaline cartilage [3]. This is why microfracture is only suitable for smaller defects (measuring < 3 cm^2^) [21]. However, the outcomes after 2–5 years after microfracture reveal alleviated pain and higher functional knee scores (KOOS, IKDC, Lysholm). In a 10-year follow-up, Gobbi et al. reported a significant improvement in two years postoperative IKDC and Lysholm scores in patients with defect sizes of 4.01 ± 0.27 cm^2^ (Lysholm pre-surgery: 45.4 ± 3.5; two years: 90.4 ± 1.8; five years: 84.7 ± 3.6; 10 years: 77.2 ± 3.5.; IKDC pre-surgery 46.7 ± 2.9; two years: 82.7 ± 3.2; five years: 79.0 ± 4.1; 10 years: 71.5 ± 4.0). Nevertheless, significantly lower scores were documented between two years and 10 years post-surgery [21]. Orth et al. reported highly similar results in their systematic review of 1870 patients [22].

The microfracture approach is limited to small defects. Defect healing with fibrocartilage instead of hyaline cartilage impairs the knee’s ability to shift the axial loading-force onto surrounding cartilage tissue [20]. Compared to this, the minced cartilage procedure showed slightly lower IKDC scores at 24 months post-surgery (pooled total mean: 77.64 ± 14.46). However, due to small cohort size, the standard deviation seen in our results is more affected by extreme outliers and is thus broader than in the work of Gobbi et al.

The long-term progress of scores remains unclear due to (so far) missing data. However, a smaller score-decrease can be alleged due to higher probabilities of generating hyaline cartilage compared to microfracture.

There are treatment options for larger, full-thickness cartilage repair with ACI/MACI, AMIC or OATS. Defect sizes in the literature range from 3 cm^2^ to a maximum 22 cm^2^, although the latter was a very rare case [23,24,25].

Autologous chondrocyte implantation (ACI) is a two-stage method. In the first stage, a small amount of cartilage tissue with viable chondrocytes is harvested from areas not involved in weight-bearing. The gathered tissue is then sent to a laboratory that amplifies and cultures these chondrocytes [26,27]. After cell amplification, the chondrocytes are surgically implanted in the osteochondral defect. The chondrocytes incorporated in scaffolds (hydrogel or decellularized type I collagen) are a modified version of the former ACI, the matrix-assisted ACI (MACI) [28,29,30].

Migliorini et al. described functional improvement after MACI in their systematic reviews [24,31]. They observed a significant improvement in IKDC and Lysholm scores. Improvement was reported over a mean 44.3-month follow-up period (IDKC pre-surgery: 38.9 ± 9.0; last follow-up: 72.1 ± 7.9; Lysholm pre-surgery: 50.1 ± 7.0; last follow-up: 82.0 ± 8.9).

However, postoperative outcomes after the ACI and MACI procedures strongly depend on the subchondral bone’s integrity too [16]. There are concerns about the dedifferentiation of chondrocytes during the in vitro amplification process, a factor that would compromise the later implant’s efficiency [3,32]. The main disadvantages of the ACI and MACI procedures are the high scaffold cost and demanding cell-processing conditions, both of which limit the widespread use of this approach [16,28,31]. In a randomized, prospective trial, Barie et al. compared MACI with ACI plus periosteum. Clinical scores were taken 12 and 24 months post-surgery as well as 8–11 years post-surgery. IKDC scores were 69.1 ± 23.6 at 12 month and 65.1 ± 29.4 at 24-month [33]. In direct comparison, the short-term scores of IKDC of our analysis demonstrated higher scores at the same point of time.

For defects exceeding 2 cm^2^ but between 2 and 8 cm^2^, autologous matrix-induced chondrogenesis (AMIC) and OATS are well-established single-step procedures [23,25,33]. Kim et al. compared the clinical and radiological outcomes of patients who underwent microfracture and AMIC. Their two-year follow up analysis demonstrated a significantly higher mean rise in IKDC and Lysholm scores following AMIC than in patients who underwent microfracture [34]. In a meta-analysis Steinwachs et al. demonstrated significant IKDC improvement in five studies after 24 months compared to baseline score. Results ranged from 65.4 to 88.0. With a pooled mean score of 77.64 ± 14.46, the IKDC scores for minced cartilage are within the same range.

Similar to the improvement and adaptation of cell application in ACI and later MACI, the minced-cartilage procedure has been optimized as well [11]. Harvesting cartilage tissue is performed using loose intra-articular cartilage fragments. However, suitable alternative options are cell harvesting from non-weightbearing sides or from defect zone edges during preparation. The latest data on tissue quality reveal topographical differences in collagen and aggrecan expression across typical cartilage-harvest sites in the knee: the lowest expression was reported from typical cartilage biopsy regions, highest collagen 1 and expression 2 at the patellofemoral joint [35]. This supports the findings of Aurich et al., who reported superior redifferentiation potential of cartilage harvested from the edge of the defect zone [36,37,38].

The fixation of fragmented cartilage onto the defect zone was first reported as an open, non-arthroscopic procedure in conjunction with using fibrin glue, which remains a widely used fixation method [9,37,38,39,40,41]. To establish healthy cartilage boarders surrounding the cartilage defect zone, the area is sharply circumcised with a scalpel and finally curettaged. Cartilage chips drawn from either the intercondylar notch or defect boarder zones is then fragmented and filled into the defect. Some authors report the pre-application of fibrin glue before and application of cartilage fragments [11]. Depending on the size and site of the defect, e.g., condylar or retropatellar, the procedure can even be fulfilled arthroscopically.

To optimize the mechanical stability of transferred minced cartilage and its adherence to the defect zone, closing the defect with a collagen membrane or specialized scaffold (CAIS scaffold: absorbable copolymer foam of 35% polycaprolactone and 65% polyglycolic acid, reinforced with polydioxanone mesh) can enhance initial stability [4,11,42,43]. The use of platelet rich plasma (PRP) in combination with autologous thrombin as an autologous sealant is currently discussed. However, the use of PRP in cartilage repair shows controversial results [11].

A comparison of inter-method results after 24 months showed improved IKDC scores in all the reported procedures. However, concerning the quality of mechanical cartilage, ACI/MACI and the minced-cartilage procedure enable the formation of hyaline-like cartilage. Fibrocartilage is more likely to form after microfracture formation: over the short term, this may alleviate symptoms and pain, but because of its rigidity, the knee’s condition worsens over the long term [5,6,21]. Measured against ACI/MACI data, IKDC’s improvement nearly equals that of the minced cartilage procedure. However, the defects associated with minced cartilage were smaller

### Limitations

There are limitations to this study inherent in the type and small number of publications we included and in our search algorithm. Our search strategy followed an English search algorithm. Potentially suitable publications in other languages were not considered. The risk of publication bias is imminent because only published articles were included. To minimize this kind of bias, the CochraneLibrary^®^ was scanned for clinical trials, but we detected no relevant findings, as the results of several ongoing trials have not been published yet.

All of the publications we included entail the risk for selection, detection, and reporting bias. To exclude methodologically inadequate studies, we focused on bias-assessment as conducted by ROBINS-I and mCMS. There was no critical risk of bias in any included study.

## 5. Conclusions

Minced cartilage is a promising method for repairing small cartilage defects as it is a single-step procedure and the production of transplant-capable material is less complex. At 12 and 24 months, clinical scores showed nearly same results as seen in commonly used cartilage repair techniques, e.g., ACI, MACI microfracture, or OATS. We thus believe it to be a very promising alternative to standard ACI/MACI, especially if these procedures are not available. There is, however, little long-term data to back up this claim at the moment, as we only have access to outcomes two years post-surgery and only a small number of studies and patients. Further research should comprise larger cohorts and the evaluation of long-term results to effectively compare results over more than five years.

## Figures and Tables

**Figure 1 jpm-12-01923-f001:**
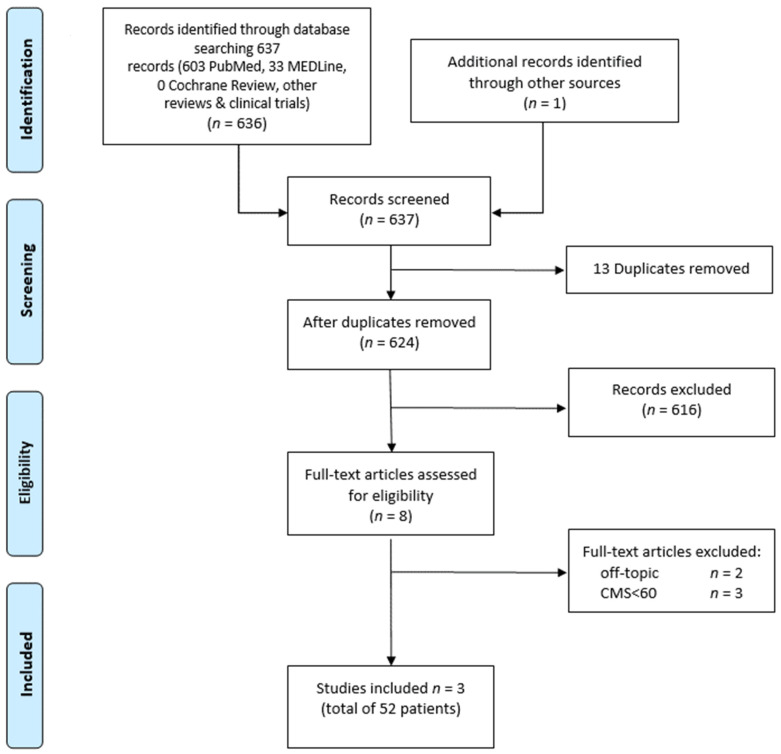
Overview of study selection process according to PRISMA guidelines.

**Figure 2 jpm-12-01923-f002:**
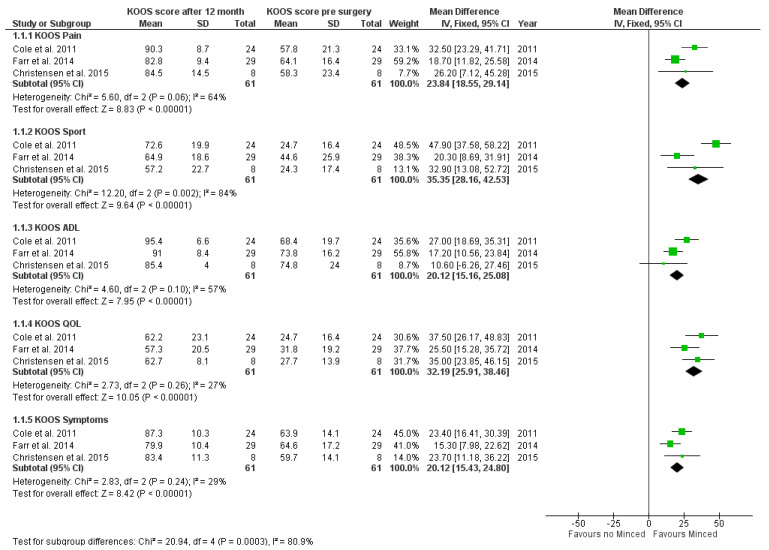
Forest plot estimating mean differences for 12-month time interval for each KOOS score subgroup.

**Figure 3 jpm-12-01923-f003:**
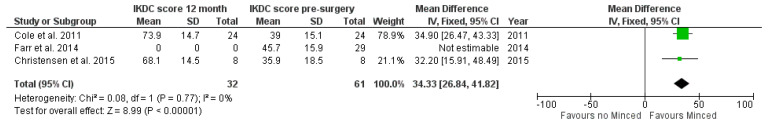
Forest plot estimating mean differences for 12-month time interval for IKDC score.

**Figure 4 jpm-12-01923-f004:**
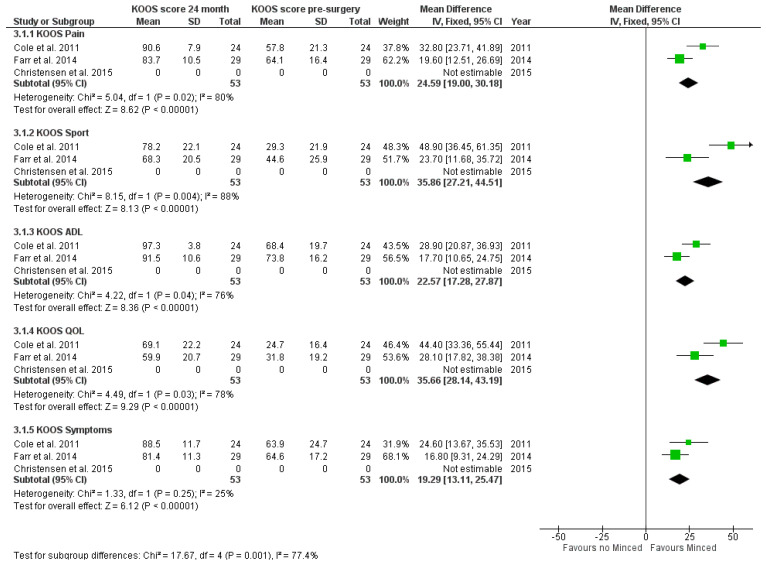
Forest plot estimating mean differences for 24-month time interval for each KOOS score subgroup.

**Figure 5 jpm-12-01923-f005:**
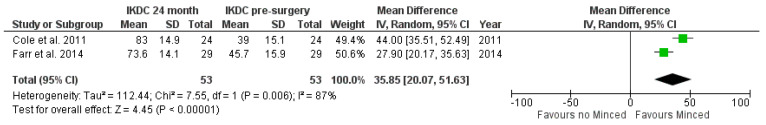
Forest plot estimating mean differences for 24-month time interval for IKDC score.

**Table 1 jpm-12-01923-t001:** Risk of bias assessment with ROBINS-I tool implicating moderate risk of bias in included studies.

*Risk of Bias Preintervention and at-Intervaention Domains*					Risk of Bias Post-Intervention Domains			
*study*	Bias due to confounding	Bias due to selection of participants into study	Bias in classification of intervention	Bias due to deviation from intended intervention	Bias due to missing data	Bias in measurement of outcome	Bias in the selection of reported outcome	Overall assessment of bias
*Christensen*	Low	Low	Moderate	Low	moderate	moderate	Low	moderate
*Farr*	Low	Low	Low	low	moderate	moderate	low	moderate
*Cole*	low	low	low	low	low	low	low	low
*Buckwalter*	serious	moderate	Low	low	moderate	moderate	low	serious
*Massen*	serious	serious	low	Low	low	moderate	low	serious

Key: 
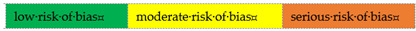
.

**Table 2 jpm-12-01923-t002:** Detailed overview of individual study mCMS scores.

Modified Coleman Methology Score					
	Christensen et al.	Cole et al.	Farr et al.	Buckwalter et al.	Massen et al.
**PART A**					
Study size	0	0	0	0	0
Mean follow up	0	0	0	0	0
Percent of Patients with follow up	3	5	5	0	5
Numbers of interventions per group	10	10	10	10	10
Type of study	10	15	10	10	0
Diagnostic certainly	5	5	5	5	5
Description of surgical technique	5	5	5	5	5
Definition of postoperativ rehabilitation	5	5	5	5	0
**PART B**					
Outcome criteria	10	10	10	8	7
Procedure of assaying outcomes	11	11	11	11	11
Description of subject selection Prozess	5	10	5	5	5
	64	76	66	59	48

**Table 3 jpm-12-01923-t003:** Demographic data from the selected studies including surgical procedure.

Study	Number of Patients	Age (y)	Male-Female	Defect Location	Defect Size (cm2)	Defect Type	Operation Method
Christensen	8 Patients	32 +/− 7	female 3 male 5	Femur condyle 7 Trochlea 1	3.1 (1.5-4.7)	Osteochondritis dissecans	Minced cartilage with fibrin glue, additional bine reconstrution
Cole	20 Patients 24 Lesions	32.7 +/− 8.8	female 6 male 14	Femur condyle 14 Trochlea 10	2.75 +/− 1.5	ICRS grade III 20 ICRS grade IV 4	Minced cartilage with fibrin glue and CAIS Scaffold
Farr	25 Patients 29 Lesions	37 +/− 11.1	femlae 7 male 18	Femur condyle 18 Trochlea 11	2.7 +/− 0.8	ICRS grade III 23 ICRS grade IV 6	Juvenile minced cartilage allograft fixed with fibrin glue

CAIS: Cartilage Autograft Implantation System.

## Data Availability

All data are within the manuscript, not applicable.

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
