# Peer review of "Minced Cartilage Is a One-Step Cartilage Repair Procedure for Small Defects in the Knee—A Systematic-Review and Meta-Analysis"

_jpm, 2022, doi:10.3390/jpm12111923_

Round 1

Reviewer 1 Report

1.      Please make sure the authors have been followed PRISMA 2020.

2.      Please end your abstract with a "take-home" message.

3.      Put the keywords in a new order based on alphabetical order.

4.      It is unclear whether the author's something new in this review. According to evaluation, several published studies by other researchers in the past adequately explain the issues you made in the present paper. Please be careful to highlight in the introduction section anything really innovative in this work.

5.      In the last paragraph of the introduction, please specifically explain the objective of the present article.

6.      The authors need to mention replacement surgery that related to cartilage damage. The introduction and/or discussion part of an article should contain this crucial information. In addition, to support this explanation, the MDPI-suggested reference should be included as follows: Jamari, J.; Ammarullah, M. I.; Santoso, G.; Sugiharto, S.; Supriyono, T.; van der Heide, E. In Silico Contact Pressure of Metal-on-Metal Total Hip Implant with Different Materials Subjected to Gait Loading. Metals (Basel). 2022, 12, 1241. https://doi.org/10.3390/met12081241

7.      For database source, the authors needs to adopt three main databases, there are Scopus, PubMed, and Web of Science.

8.      Please include the limitation of the present review, it is missing.

9.      In the conclusion, please explain the further research.

10.   The reference should be enriched with literature from the last five years. MDPI reference is strongly recommended.

11.   The manuscript needs to be proofread by the authors since it has grammatical and language issues.

12.   A graphical abstract is suggested to be included in the submission after peer review.

Author Response

Dear Reviewer,

first of all, we want to thank you for your detailled review.

The Manuscript has been checked by an native speaker/proof reader and should adequate in language and style.

  1. Please make sure the authors have been followed PRISMA 2020.

A PRISMA 2020 Check-lsit has been attached as complemantary file.

  1. Please end your abstract with a "take-home" message.

we added a take-home message to the abstract; line 32-34

  1. Put the keywords in a new order based on alphabetical order.

Keywords have been placed alphabetical

  1. It is unclear whether the author's something new in this review. According to evaluation, several published studies by other researchers in the past adequately explain the issues you made in the present paper. Please be careful to highlight in the introduction section anything really innovative in this work.

As there is no paper summarizing data as a meta-analysis our study presents the first that actually synthesizes current data. According to this, we adapted the abstract; line 16-18

  1. In the last paragraph of the introduction, please specifically explain the objective of the present article.

The objective has been pointed out more specifically; line 64-66

  1. The authors need to mention replacement surgery that related to cartilage damage. The introduction and/or discussion part of an article should contain this crucial information. In addition, to support this explanation, the MDPI-suggested reference should be included as follows: Jamari, J.; Ammarullah, M. I.; Santoso, G.; Sugiharto, S.; Supriyono, T.; van der Heide, E. In Silico Contact Pressure of Metal-on-Metal Total Hip Implant with Different Materials Subjected to Gait Loading. Metals (Basel). 2022, 12, 1241. https://doi.org/10.3390/met12081241

As this paper is thematically about cartialge repair techniques, cartialge replacement surgery would be off topic. Eventhough, the article you recommended is very interesting, we will not add cartilage repacement surgery thematically in our manuscript.

  1. For database source, the authors needs to adopt three main databases, there are Scopus, PubMed, and Web of Science.

Databases have been added, as MEDLine and EMBASE comprise results of Scopus, we did not add Scopus specifically; line 72

  1. Please include the limitation of the present review, it is missing.

The missing limitations section has been added; line 265-275

  1. In the conclusion, please explain the further research.

an excerpt of further research has been added in the conclusion; line 284-286

  1. The reference should be enriched with literature from the last five years. MDPI reference is strongly recommended.

the reference list was scanned and we tried to supplement/exchange for suitable "newer" studies when applicable.

  1. The manuscript needs to be proofread by the authors since it has grammatical and language issues.

done

  1. A graphical abstract is suggested to be included in the submission after peer review.

done

Reviewer 2 Report

1. Kindly state whether the protocol for this review is registered. It is mandatory. Please specify the registration number and access link.

2. Kindly adhere to PRISMA statement. State this in the methodology. Please provide a completed PRISMA checklist as a supplementary material.

3. State the eligibility of included studies in the following format: PICO - Patient, intervention, control, and outcome.

4. Mention the search strategy used for obtaining the studies.

5. Mention how many authors were involved in searching the appropriate articles, data extraction, etc.

6. State how were the conflicts resolved in case of disagreements.

7. What is CMS in Figure 1.

8.Please include a comprehensive legend to all the figures.

9. Add meta-analysis in the title.

10. Mention the details of how meta-analysis was done in the statistical analysis section.

11. Why was fixed-effects model used with I2 of 84%? Not acceptable.

12. Grade the findings mentioning their strength of recommendations.

Author Response

1.Kindly state whether the protocol for this review is registered. It is mandatory. Please specify the registration number and access link

Protocoll has been submitted to PROSPERO, we added this information to the "methods" part; As the registration process is still ongoing, it does currently not include a regsitrationnumber; line 69-70

2. Kindly adhere to PRISMA statement. State this in the methodology. Please provide a completed PRISMA checklist as a supplementary material.

PRISMA-Checklist has been added as supplementary material

3. State the eligibility of included studies in the following format: PICO - Patient, intervention, control, and outcome.

The presenting format has been changed: line 86-87

4. Mention the search strategy used for obtaining the studies.

The search strategy is depicted in a table showing combinations of terms and number of results

5. Mention how many authors were involved in searching the appropriate articles, data extraction, etc.

There were two authors involved in scanning articles and data extraction; A.F. and T. Y., it is already stated in the methods section, line 71

6. State how were the conflicts resolved in case of disagreements.

Due to an objectifiable extraction/collection process with the Coleman Methodology score and ROBINS-I tool, disagreements didn't come up

7. What is CMS in Figure 1.

mCMS or CMS is the abbreviation for "modified Coleman Methodology score"; an explanation of the abbreviation is given in line 87-88

8.Please include a comprehensive legend to all the figures.

done

9. Add meta-analysis in the title.

done

10. Mention the details of how meta-analysis was done in the statistical analysis section.

As RevMan5 was used for analysis, the installed subroutines combined and analyzed data. We put this into account; line 104-105

11. Why was fixed-effects model used with I2 of 84%? Not acceptable.

totally right, we changed the figure to an random effects model

12. Grade the findings mentioning their strength of recommendations.

The central finding of this study was an improvement of clinical scores nearly equal to "standard cartilage repair techniques"; Due to a small cohort and only short-term results it is not possible to give clear recommendations; We stated this in our conclusion; line 339-342

Round 2

Reviewer 1 Report

Reviewers greatly appreciate the efforts that have been made by the author to improve the quality of their articles after peer review. I reread the author's manuscript and further reviewed the changes made along with the responses from previous reviewers' comments. Unfortunately, the authors failed to make some of the substantial improvements they should have made making this article not of decent quality with biased, not cutting-edge updates on the research topic outlined. In addition, the author also failed to address the previous reviewer's comments, especially on comments number 4 (it needs to evidence with presentation of literature searching based on Scopus, PubMed, and Web of Sciecne, 6 (suggested literature not incorporated), and 7 (Change MEDLine and Embase with Scopus), . With all due respect, the reviewer opposed this article to be published and must be rejected. Thank you very much for the opportunity to read the author's current work.

Author Response

Dear Reviewer,

we thank you for your efforts and advice to optimize our manuscript. However, in some points we clearly disagree with your suggestions (suggestion #6), as artificial cartilage replacement therapy is off topic in a paper adressing biological treatment options for cartilage repair.

Reviewer 2 Report

Actually, the protocol is considered to be registered only if you have the registration number.

Author Response

Dear Reviewer,

until now the processing at PROSPERO is still ongoing. As the manuscript is not an UK submission the process is suspected to be ab bit longer.

"There is currently a very high demand for registration, we will aim to
respond within 10 working days for UK submissions. During this time
you may continue working on your review.  You can be reassured that
the team are working particularly hard to process records as quickly
as is possible.
"

As soon as the registration will be done, the registration-ID will be added to the manuscript.